# Health economic assessment of a scenario to promote bicycling as active transport in Stockholm, Sweden

Hedi Katre Kriit,[1] Jennifer Stewart Williams,[2] Lars Lindholm,[2] Bertil Forsberg,[1] Johan Nilsson Sommar [1]

[1]Sustainable Health, Department of Public Health and Clinical Medicine, Umeå University, Umeå, Sweden
[2]Department of Epidemiology and Global Health, Umeå University, Umeå, Sweden

**Correspondence to**
Hedi Katre Kriit;
hedi.kriit@umu.se

## ABSTRACT

**Objectives** To conduct a health economic evaluation of a proposed investment in urban bicycle infrastructure in Stockholm County, Sweden.

**Design** A cost-effectiveness analysis is undertaken from a healthcare perspective. Investment costs over a 50-year life cycle are offset by averted healthcare costs and compared with estimated long-term impacts on morbidity, quantified in disability-adjusted life years (DALYs). The results are re-calculated under different assumptions to model the effects of uncertainty.

**Setting** The Municipality of Stockholm (population 2.27 million) committed funds for bicycle path infrastructure with the aim of achieving a 15% increase in the number of bicycle commuters by 2030. This work is based on a previously constructed scenario, in which individual registry data on home and work address and a transport model allocation to different modes of transport identified 111 487 individuals with the physical capacity to bicycle to work within 30 min but that currently drive a car to work.

**Results** Morbidity impacts and healthcare costs attributed to increased physical activity, change in air pollution exposure and accident risk are quantified under the scenario. The largest reduction in healthcare costs is attributed to increased physical activity and the second largest to reduced air pollution exposure among the population of Greater Stockholm. The expected net benefit from the investment is 8.7% of the 2017 Stockholm County healthcare budget, and 3.7% after discounting. The economic evaluation estimates that the intervention is cost-effective and each DALY averted gives a surplus of €9933. The results remained robust under varied assumptions pertaining to reduced numbers of additional bicycle commuters.

**Conclusion** Investing in urban infrastructure to increase bicycling as active transport is cost-effective from a healthcare sector perspective.

## Strengths and limitations of this study

► This transport mode shift scenario is based on registry data and reflects potential outcomes from increased bicycling in Stockholm County.
► Due to data limitations, it was not possible to model possible changes in population health in the absence of the intervention.
► Investment costs are compared with the expected reduction of chronic disease-related healthcare costs due to the transport mode shift.
► Secondary data were used to estimate the lifetime costs of chronic diseases.

## BACKGROUND

Four chronic illnesses—cardiovascular disease, cancer, type 2 diabetes and respiratory disorders—account for at least 60% of global deaths. Several known modifiable lifestyle behaviours, such as tobacco smoking, unhealthy diets, harmful alcohol use and physical inactivity, impact on these chronic illnesses through associations with risk factors, such as obesity, hypertension and high blood sugar levels.[1–6] Physical inactivity is the fourth most important risk factor for premature mortality in the European Union after tobacco smoking, high blood pressure and overweight.[5]

There is now widespread evidence of the importance of physical activity in promoting and maintaining good health.[3 5 6] Even the relatively inactive can benefit from marginal increases in physical activity.[7] Since 2010, when the first official set of global recommendations for physical activity were launched by the World Health Organization (WHO),[3] policies to address insufficient physical activity have been implemented in over half WHO Member States.[6]

Currently, half of the world's population live in urban environments and this proportion is expected to reach 60% by 2030.[8 9] Participation in physical activity can be difficult in cities where the heavy use of motorised transportation is estimated to be responsible for 70% of environmental urban air pollution and 13% of global green gas emissions.[10] In 2010, road transport contributed 40% of the economic costs of premature deaths from air pollution in the WHO European Region.[11]

Active transport through, for example, bicycling as a way of commuting to work has

been suggested as a way of increasing physical activity and reducing motorised traffic emissions in urban areas. However, the health gains for those switching from car to bike transport in heavy traffic areas are offset by the increased risk of serious injury and greater exposure to poor air quality. Furthermore, the distribution of costs and benefits is not equally spread across socioeconomic and demographic groups.[7 12 13]

In a study of the cost-effectiveness of interventions to encourage more physical activity in the Australian population, Cobiac et al reported that a package of physical activity interventions that include active transport could yield both public health benefits and health sector cost savings.[14] Another study in Australia estimated that healthcare costs would be reduced by $A1.12 per bicycled kilometre.[15] Studies using previously published disease-specific healthcare costs have estimated significant cost reductions based on hypothetical scenarios of increased active commuting in the UK[16] and Italy.[17] For example, Jarret et al estimated that the National Health Service could save roughly £17 billion within 20 years due to reduced prevalence of seven non-communicable diseases (NCDs) resulting from increased walking and bicycling.[16] A number of economic assessments have suggested that travel-related/transport-related interventions are cost-effective in terms of costs per healthy life-year gained in adult populations.[7 18] Although, not without methodological challenges, there is a need to evaluate the costs and benefits of interventions that promote physical activity through active transport from a health sector perspective.[19 20]

In 2015, NCDs accounted for €115 billion in healthcare costs in European countries.[21] Strategies that address physical inactivity can deliver public health gains through the prevention of chronic diseases. Active transport strategies involving a shift from motor transport can also have wider benefits for the population through improvements in air quality. Sound economic evaluations are needed to build an evidence base in this area.[2]

This paper presents a health economic evaluation of a proposed investment in urban bicycle infrastructure in Stockholm, Sweden, from a healthcare sector perspective. The long-term change in outcomes and healthcare costs are quantified within a scenario of increased commuting by bicycle and cost-effectiveness is derived in relation to the planned investment.

## METHODS

### Evaluation scenario

As part of its 2018–2030 plan, the Municipality of Stockholm (population 2.27 million) committed funds for bicycle path infrastructure with the aim of achieving a 15% increase in bicycle commuters by 2030. Based on the previously developed transport shift scenario, it is estimated that 111 487 current car drivers could potentially switch to bicycle commuting between home and work. Thereafter, in the economic evaluation, it is assumed that yearly investment costs lead to a proportional amount of new bicyclists from the estimated number of potential converters. Health impacts, attributed to increased physical activity, differential exposure to air pollution and traffic accidents, and changes in healthcare costs are evaluated for potential bicycle commuters, current bicycle commuters and the general population. The timeline assumes that it takes 12 years of investment for all estimated potential bicyclists to shift their mode of transport and health impacts are calculated for a 50-year life length of the investment in bicycle path infrastructure.[22] Yearly health impacts are compared, ceteris paribus, with current 2017 register data on disease incidences that were assumed to be constant during the 50 years of assessment. The corresponding estimated lifetime healthcare costs were calculated and used as a comparator for cost-effectiveness analysis.

### Population base

The study population is based on a scenario constructed for a previous study in which data on geolocation, demography and commuting modes by origin and destination zone were referenced.[22] Home and workplace addresses were retrieved from the ASTRID database, which has been collecting individual-level data on the Swedish population for over 50 years.[23] Using the LuTrans transport model,[24] each individual within ASTRID was allocated to different modes of commuting based on travel survey information and registry data on car ownership. Regression functions were used to model the probability for each mode of travel between small statistical areas of Stockholm County. A total of 111 487 individuals from the Stockholm County population register who currently drive a car to work were identified as potential bicycle commuters based on geographic distance from home to work and expected bicycle speed based on age and sex. Eligibility required an estimated bicycle commute of 30 min or less each way.[22 25]

### Estimating health outcomes

Sommar et al[26] estimated changes in morbidity and mortality associated with the above chronic diseases and traffic accidents under the scenario, for potential bicycle commuters, current bicycle commuters and the general population. First, estimated health impacts due to increased physical activity among potential bicycle commuters were quantified in terms of decreased risk of type 2 diabetes, myocardial infarction (MI), stroke, dementia, colon cancer, breast cancer and heart failure. Second, change in air pollution exposure is assumed to benefit current bicycle commuters and the general population and disadvantage additional bicycle commuters in terms of risk for asthma, lung cancer, type 2 diabetes, stroke and MI.[26] Third, health effects of injuries attributable to traffic accidents under this scenario of 111 487 potential bicycle commuters are also estimated.[27] Estimated disability-adjusted life years (DALYs) were calculated for each area of health impact based on the disability weights reported by the WHO Global Burden

of Disease 2004 updates and expected changes in disease incidence. The duration of the disease was calculated as the expected life length after the first disease incidence. Life table calculations using mortality data for Stockholm County were used to calculate expected remaining life years based on age and gender.[26]

### Estimating cost-effectiveness

Information on the budget allocation for the bicycle pathways was obtained from the Stockholm Traffic Planning Office.[22 28] The expected life cycle for new bicycle paths constructed in 2018 is 50 years.[29]

The investment period of 2018–2030 was used as a build-up time to reach the estimated amount of additional bicyclists. We estimate that yearly investment costs will vary due to political and other circumstances. The number of additional bicyclists estimated to start active commuting in a given year is a proportion of the estimated potential bicycle commuters, based on that year's investment allocation (approximately €15 million per year). Health impacts were simultaneously estimated consistent with the proportion of additional bicyclists in a given year. After 2030, an estimated 111 487 additional bicyclists were expected to commute. The number of DALYs attributable to this transport mode change was modelled for the remaining years. Long-term healthcare costs (added and averted) are estimated in accordance with the change in health outcomes attributed to: (1) increased physical activity by potential bicycle commuters; (2) air pollution change for potential bicycle commuters; (3) air pollution change for current bicycle commuters; (4) air pollution change for the general population and (5) traffic accidents incurred by potential bicycle commuters. These costs are given both in absolute terms and as the proportion of the 2017 Stockholm Municipality health budget.[30]

Health outcomes are calculated for the 50-year period, 2018–2068, taking the time interval between exposure and realised health benefits into account. It is estimated for physical activity exposure that it takes 2 years for 50% of the health benefits and 6 years for 100% of the health benefits to be realised for both MI and stroke,[31] 17 and 20 years, respectively, for 50% and 100% of the health benefits to be realised for dementia, breast and colon cancer,[31] and 3.2 and 8 years, respectively, for 50% and 100% of the health benefits to be realised for type 2 diabetes.[32] The impacts of air pollution and traffic injuries were both assumed to occur during the current year. The mode shift from car to bicycle was assumed to be the only change in the exposure.

This cost-effectiveness analysis adopts the Consolidated Health Economic Evaluation Reporting Standards (CHEERS) guidelines for health economic modelling.[33] Costs are reported in euros (2017). Figure 1 shows the steps undertaken. The cost-effectiveness assumed that healthcare costs would be reinvested yearly, and therefore discounting was not used. Costs are, however, also reported with a 3% discount rate.

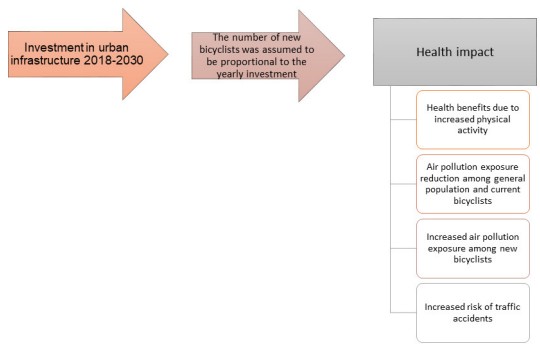

**Figure 1** Flow chart for estimated chain of events as the base for conducting the cost-effectiveness analysis.

### Sensitivity analyses

Sensitivity analysis is undertaken for the same investment costs assuming that numbers of potential bicycle commuters are as few as 25% of the estimated potential bicyclists. In addition, the sensitivity to a reduced health impact and healthcare cost is provided. Results were reported in terms of its effect on the cost-effectiveness ratio (CER).

### Estimating long-term healthcare costs

The literature was used to inform estimates of long-term healthcare costs for type 2 diabetes, MI, stroke, heart failure, dementia, colon cancer, breast cancer, lung cancer and asthma (table 1). For comparability with Sweden, only papers reporting studies in high-income countries, in which healthcare costs were met by public funds, were sought. Inclusion required that study estimates were based on administrative data recording healthcare costs matched to clinical diagnoses. The search was conducted in *Google Scholar*, *PubMed* and *Web of Science* between 1995 and 2018. Search terms included the health condition of interest and *lifetime/long-term costs*. As no studies were found that specifically reported lifetime healthcare costs for asthma, MI or cancer outcomes, these estimates are based on 1-year average healthcare costs for asthma, 6-year average for MI and 5-year disease costs for cancer.

Estimates of the costs for bicycle-related injuries were derived from a study conducted in the Netherlands, which used national registry and hospital data covering the period 1998–2012.[34] Reported average cost per injury was €2765.

## RESULTS
### Investment cost

The net infrastructure cost, excluding communication and maintenance costs, is €101 million. Assuming a 50-year life cycle for the bicycle path, a €2 million annual investment gives an average annual cost of €908 per converter.

### Changes in healthcare costs

The expected healthcare costs assuming the current burden of included diseases during the 50-year life span

**Table 1** Chronic disease treatment costs per case derived from the literature used to calculate healthcare costs

| Disease | Sample size | Sample population | Data collection | Perspective | Method | Costs per case | Disease duration | Reference |
|---|---|---|---|---|---|---|---|---|
| Type 2 diabetes | 1000 | Newly diagnosed type 2 diabetes | 2009–2010 | Healthcare | Markov modelling | €83 1171 | Lifetime | Zhou et al[54] |
| Stroke | 9064 | All first stroke cases registered in the national register | 2009 | Health and societal | Aggregated costs | €74 138 | Lifetime | Ghatnekar et al[55] |
| Dementia | 111 811 | Medicare beneficiaries≥65 years | 1997–2005 | Healthcare | Cohort-based simulations | €13 767 | Lifetime | Yang et al[56] |
| Myocardial infarction | 41 210 | Acute myocardial infarction incidences in Alberta, Canada | 2004–2014 | Healthcare | Cohort-based aggregated costs | €17 784 | 6 years | Tran et al[57] |
| Heart failure | 1054 | Olmsted County residents who fulfilled the special criteria's established for heart failure | 2006 | Healthcare | Tiangand-Huang method | €113 996 | Lifetime | Dunlay et al[58] |
| Breast cancer | 77 009 | Cohort of elderly Medicare, newly diagnosed cancer patients | 1998–2004 | Healthcare | 5-year aggregated costs | €21 181 | 5 years | Yabroff et al[59] |
| Colon cancer | 44 838 | Cohort of elderly Medicare, newly diagnosed cancer patients | 1998–2004 | Healthcare | 5-year aggregated costs | €42 461 | 5 years | Yabroff et al[59] |
| Lung cancer | 54 665 | Cohort of elderly Medicare, newly diagnosed cancer patients | 1998–2004 | Healthcare | 5-year aggregated costs | €43 219 | 5 years | Yabroff et al[59] |
| Asthma | 49 668 Control group: 248 340 | Patients with ICD-10-GM asthma code extracted from German sickness fund* | 2010 | Healthcare | Aggregated costs | €49 537 | Lifetime, calculated from a reported one year estimate and thereafter multiplied by the expected life length[60] with the disease | Jacob et al[61] |

*ICD-10-GM –The International Statistical Classification Of Diseases And Related Health Problems, 10th revision, German Modification.

**Table 2** Estimated healthcare costs assuming the 2017 disease burden, changed disease burden as a result of increased bicycling, and estimated difference in healthcare costs

| Exposure | Healthcare costs with the current burden of disease | Healthcare costs in the scenario | Estimated change in healthcare costs in the scenario | Percentual reduction in healthcare costs | Estimated change in DALYs in the scenario |
|---|---|---|---|---|---|
| Physical activity | €2 073 803 338 | €1 511 162 940 | €562 640 398 | 27.13 | 35 060 |
| Air pollution change among the general population | €46 283 254 008 | €46 228 719 501 | €54 534 507 | 0.12 | 21 975 |
| Air pollution change among additional bicyclist | €1 885 540 341 | €1 899 564 512 | €−14 024 171 | −0.74 | −7 287 |
| Air pollution change among current bicyclists | €874 825 533 | €874 566 906 | €258 627 | 0.03 | 131 |
| Traffic injuries* | | | €−88 244 441 | | −8 210 |
| Total | €46 283 254 008 | €45 768 089 088 | €515 164 920 | 1.01 | 41 669 |

*Current costs related to traffic accidents were not available.
DALYs, disability-adjusted life years.

of the investment was €46 billion (table 2), and €24 billion after discounting (table 3).

The fall in chronic disease cases attributable to increased physical activity among additional bicycle commuters is estimated to save €562 million, or €241 million after discounting (3%), giving an average annual return of €11.24 million (figure 2).

Traffic generated air pollutants were measured in particle mass concentrations.[35] Lower healthcare costs attributed to decreased air pollution exposure among current bicycle commuters and the general population are estimated to save €1.2 million annually. This is offset by increased healthcare costs of €0.28 million for additional bicycle commuters who are estimated to experience increased exposure to air pollution (figure 2).

The annual additional healthcare costs for injuries sustained as a result of the increased accident risk among additional bicycle commuters is estimated at €1.96 million.

During the life span of the investment, the estimated total healthcare costs for the included diseases was reduced by 1.1%. This corresponds to 8.7% of the healthcare expenditure that the Stockholm Council planned for the year 2017, or 3.7% after discounting.

## Costs and benefits
Costs for the given scenario are compared with no assumed change in bicycle commuting. For each area of impact, the estimated change in healthcare costs and DALYs were calculated (table 2). CER is calculated by

**Table 3** Discounted costs assuming the 2017 disease burden, changed disease burden as a result of increased bicycling, and estimated difference in healthcare costs

| Exposure | Healthcare costs with current burden of disease | Estimated change in healthcare costs in the scenario | Percentual reduction in healthcare costs |
|---|---|---|---|
| Physical activity | €1 092 130 168 | €241 629 107 | 22.12 |
| Air pollution change among the general population | €24 257 656 022 | €25 962 558 | 0.11 |
| Air pollution change among additional bicyclist | €988 236 242 | €−6 687 730 | −0.68 |
| Air pollution change among current bicyclists | €458 507 452 | €108 504 | 0.02 |
| Traffic injuries* | | €−42 117 426 | |
| Total | €24 257 656 022 | €218 895 013 | 0.82 |

*Current burden costs related to traffic accidents were not available.

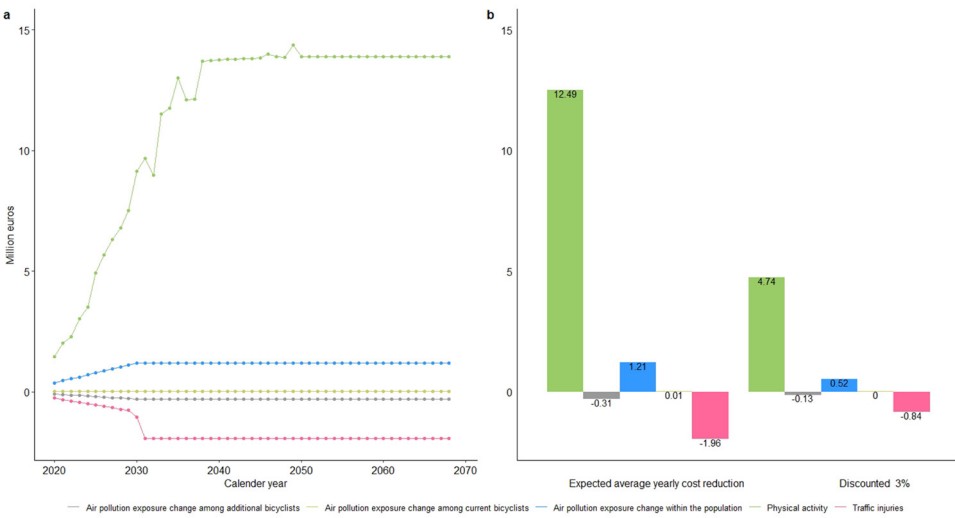

**Figure 2** Estimated yearly expenditure averted (in millions) in the healthcare sector due to increased physical activity, change in air pollution concentrations and risk of traffic injuries.

subtracting investment costs from the total estimated healthcare savings and dividing by the estimated reduction of DALYs. Based on our estimates, the investment is cost-effective, averting one DALY gives a surplus of €9933.

### Sensitivity analysis
The scenario identified 111 487 additional bicycle commuters. The sensitivity of the estimates was assessed by assuming fewer potential bicycle commuters. If only 25% of the 111 487 potential bicycle commuters were to change their mode of transport, the CER would still result in a surplus of €2643/DALY averted (figure 3). Even under the assumptions of a reduced health impact and lower expected reductions in healthcare costs, the investment remains highly cost-effective.

### DISCUSSION
This is the first health economic evaluation of this type to use an evidence-based scenario that includes information on geolocation, demography and commuting modes by origin and destination zone drawn from a

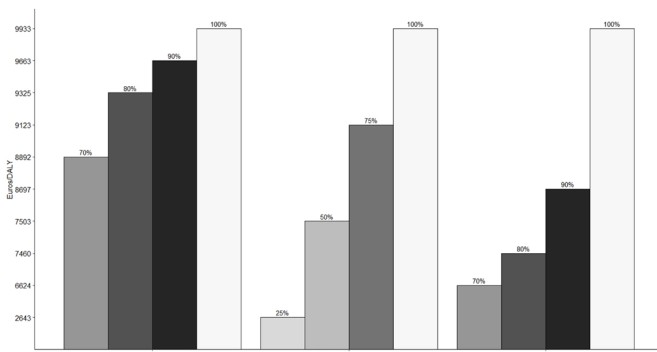

**Figure 3** Sensitivity analysis of the CER considering number of obtained bicyclists, health impact and healthcare costs. CER, cost-effectiveness ratio; DALY, disability-adjusted life year.

population register. Investment costs were compared with estimated long-term healthcare costs and savings attributed to increased physical activity, change in air pollution exposure and accident risk. To the best of our knowledge, this is also the first economic evaluation to include morbidity health impacts in all three areas of impact. The investment is cost-effective from a healthcare sector perspective.

### Physical activity
The largest reduction in healthcare costs is attributed to increased physical activity. Previous health economic assessments using hypothetical scenarios in Europe have also demonstrated cost savings in the healthcare sector attributed to physical activity.[16 17] In the USA, Grabow *et al*[36] showed that in a population of 2 million across 11 metropolitan areas, if one-half of the commutes, of less than 8 km round trip, were made by bicycle instead of car, mortality would be reduced by 700 premature deaths. Based on the US statistical value of life, this is equivalent to US$3.8 billion (2010 values) annually or US$1900 per individual.[36] In our study, the estimated cost saving attributed to reduced morbidity from increased physical activity is €562 million or €100 per annum per additional bicycle commuter.

The benefits from different types of physical activity are variable and not necessarily linear. In their systematic review and meta-analysis of 22 cohort studies including over 977 000 people, Woodcock *et al*[37] concluded that, compared with no activity, 2.5 hours per week of moderate-intensity activity was associated with a 19% reduction in mortality risk and 7 hours per week of moderate activity reduced mortality risk by 24%. In this study, physical activity effects are reported only for bicycling although adjustments were made for other likely types of physical activity based on individuals' assumed physical capacity adjusted for age and sex.

## Air pollution

Some other health economic analyses have assessed the economic value of a reduction in morbidity and mortality due to a reduction in air pollution exposure. Lindsay et al,[38] for example, estimated the effect of reduced air pollution exposure on mortality, numbers of active days, and acute cardiac and respiratory hospital admissions by calculating cost per health event and the statistical value of life based on New Zealand willingness to pay studies. Grabow et al[36] used a BenMAP cost-benefit regulatory analysis to address the potential benefits attributable to an air pollution reduction in the general population as a result of a transport mode change from car to bicycle for trips of 8 km or less in various US cities. Based on acute hospital costs attributable to reduced automobile exhaust and increased ozone exposure, net benefits were estimated at US$4.94 billion (2010 values).

To the best of our knowledge, this economic evaluation is also the first to model the impact of a change in air pollution exposure within the general population as well as among current bicycle commuters. Reduced healthcare costs within the general population (€54 million) are offset by additional costs of €14 million among additional bicycle commuters, giving a net freed opportunity cost of €40 million within the healthcare budget over the 50-year life length of the investment.

## Injuries

In their health economic assessment of cycling promotion in Florence, Italy, Taddei et al[17] predicted decreased traffic accidents. Woodcock et al[39] modelled the health impacts of active transport visions in England and Wales and also predicted a fall in road traffic and accidents under scenarios of increased walking and cycling. However, using the same general scenario for increased bicycling in Stockholm County, Nilsson et al[27] estimated health loss through increased DALYs due to increased numbers and types of traffic accidents, being the first major study to highlight the negative impacts of increased injuries and fatalities resulting from the shift from car to bicycle. A strength of the Nilsson study, the results of which are incorporated into our analysis of the health impacts and costs of traffic injuries, is the use of nationally representative hospital and police register data on injuries occurred.

Head and spinal injuries costs were included in an analysis of the effects of increased active travel in urban England and Wales on the National Health Service.[16] However, these estimates were four times higher than the estimates used here. The current work is based on a study by Scholten et al[34] which assumed traumatic brain injuries occurring in about 9% of bicycle-related traffic accidents.

It has been claimed that 60% of the bicycle accidents occur due to poor infrastructure for bicycling in Sweden[40] and one of the aims of the Stockholm Bicycle Plan was to increase bicycle safety by building infrastructure for active commuters.[28] Improved road infrastructure has been shown to reduce bicycle fatalities by 45%[41] and it has been suggested that increasing the number of bicyclists can also have a 'safety in numbers' effect which reduces accident risk.[42] A study done by Organisation for Economic Co-operation and Development (OECD)[43] showed that Sweden had the world's safest roads with the lowest number of fatal accidents per billion-cycled kilometres compared with other European countries.

## Total economic benefits
### Healthcare costs

This study shows a net benefit corresponding to in total of 8.7% of the 2017 Stockholm County healthcare budget after 50 years. Grabow et al[36] reported a 5% reduction in healthcare cost attributable to a change in air pollution exposure and increased physical activity. In a scenario of daily bicycling of 3.4 km among urban residents in England and Wales, Jarrett et al[16] reported that the costs for England National Health Service would be reduced by 0.8% as a consequence of increased physical activity.

### Economic assessment

It is assumed that investment in physical infrastructure would lead to increased physical activity levels and flow-on cost reductions within the healthcare sector. These assumptions have been adopted by others undertaking similar economic evaluations in which investment in infrastructure to promote active transport is assumed to increase physical activity and reduce air pollution. For example, a study conducted in Dane County in the USA reported a cost–benefit ratio of 1.87 over a 10-year cycle attributed to a US$450 million investment in sidewalks.[44] Some researchers suggest that the saved opportunity costs could finance the investment. For example, a study of air quality-related and exercise-related health benefits resulting from reduced car travel in the Midwestern USA, Grabow et al,[36] concluded that the estimated reduction in healthcare costs could finance the infrastructure change within 1–10 years. In Italy, Taddei et al[17] estimated that healthcare costs saved due to a reduction in NCDs could cover 50 km of new bicycle roads within 7–10 years in Florence. According to our estimates the potential health care savings could cover the infrastructure costs of bicycle lanes in 18 years.

## Strengths and limitations
### Strengths

Many previous health economic analyses of active transport interventions have used hypothetical or proposed active travel interventions; however, the scenario in this study was constructed linking register data that included home and work address coordinates.[25] Gender-specific and age-specific expected bicycle speeds were used to assess individuals' capacity to commute by bicycle between home and work in 30 min or less.[22 25] In their systematic review of active transport interventions that include physical activity benefits, Brown et al[45] reported a limited inclusion of health effects with few studies reporting reductions in morbidity as well as mortality. Morbidity reduction is captured in the DALY weights in this study and the impacts of mortality under this

scenario have been estimated in a companion paper.[26] Additionally, the sensitivity analysis showed that the intervention is highly cost-effective even if only one-quarter of the potential bicycle commuters were to take up bicycle commuting.

## Limitations

We acknowledge the limitations in the health economic modelling. It is fair to assume that improvements in population health would still occur without the intervention or increased bicycling. However, modelling all possible behavioural changes under uncertainty and generalising these to population health outcomes in the coming 50 years would be a complex exercise beyond the scope of this study. Furthermore, it would not be meaningful to base predictions on possible future health trends for which there is a lack of reliable data. Such is the case regarding air pollution trends in Stockholm since 2010[46]. Cost-effectiveness studies facing these and similar limitations have been published previously.[47]

The health impact estimates of our economic assessment are based on the Global Burden of Disease.[48] We have not included all possible health impacts. We did not, for example, account for impacts on mental health (eg, depression) and obesity that are expected to result from increased physical activity.[49 50] A study in the USA reported that obesity reductions resulting from active transport could save US$90.93 million in healthcare costs.[12]

It is possible that health benefits and cost savings are underestimated in this study due to not fully accounting for the benefits of reduced obesity and improved well-being that may accrue over time and flow-on to the broader community. These effects would be considered in an economic evaluation conducted from a societal perspective, which is beyond the scope of the present analysis. There are other potential savings which were not included. For example, it has been shown that people value the time bicycling to work more than the time spent in motorised transport.[51] If a societal perspective had been applied, cost-effectiveness may have been higher.

Except for the effect of the mode shift towards increased bicycling, no other changes in air pollution concentrations were assumed. In terms of air pollution effects, the US Environmental Protection Agency applies segmented lag times for mortality estimates.[52] However, due to the lack of studies on lag times for air pollution effects on morbidity, it was assumed that effects occurred in the same year as exposure.

In using secondary data to estimate long-term costs for chronic NCDs, there was a lack of evidence for cancer and asthma patients. Estimates are, therefore, based on the best approximations from scientific evidence published in the peer-reviewed literature. Although it is common practice for economic evaluations to draw on evidence from previous studies, comparing evidence from studies with different methodological approaches in different settings is not without its shortcomings.[53] Studies from other high-income countries were used so that the results could be reasonably generalised to Sweden.

## CONCLUSIONS

From a healthcare perspective investing in urban infrastructure to increase bicycling in Stockholm is cost-effective in relation to the Swedish threshold value of €53 000 per DALY averted. This evaluation makes the best use of available data and calculates estimates based on a previously developed realistic scenario. More work of this type is needed to build an evidence base for decision-makers responsible for investing funds in public infrastructure.

**Acknowledgements** We thank study leaders and co-authors in the transport scenario study, in particular, Peter Schantz, Christer Johansson, Wasif Raza and Magnus Strömgren. We are grateful to the reviewers who have challenged us with constructive comments and excellent scientific feedback.

**Contributors** HKK planned the health economic study and undertook the data collection and economic analysis with input from LL. HKK wrote the first draft. JSW reviewed the overall content, provided input, and revised and re-drafted the paper in conjunction with HKK. BF advised on the planning and conduct of the study and was responsible for the overall transport scenario project of which this work was part. JS provided health impact estimates and critical input into the planning and conduct of the study, assisted with the statistical analyses and helped to revise the drafts. All the authors read and approved the final version of the manuscript.

**Funding** This work was funded by the Swedish Research Council for Health, Working Life and Welfare (grant number 2012-1296). BF received a grant (2012-1296) from FORTE (Forskningsrådet för hälsa, arbetsliv och välfärd) for the transport scenario study.

**Competing interests** None declared.

**Patient consent for publication** Not required.

**Provenance and peer review** Not commissioned; externally peer reviewed.

**Data availability statement** All data relevant to the study are included in the article or uploaded as supplementary information.

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
