## [Reviewer comments · BMJ Open]

ARTICLE DETAILS

TITLE (PROVISIONAL)	Health economic assessment of a scenario to promote bicycling as active transport in Stockholm, Sweden
AUTHORS	Kriit, Hedi; Williams, Jennifer Stewart; Lindholm, Lars; Forsberg, Bertil; Sommar, Johan

VERSION 1 – REVIEW

REVIEWER	Martin Howell University of Sydney
REVIEW RETURNED	08-Apr-2019

GENERAL COMMENTS	The economic evaluation addresses a topic of interest and relevance to prevention of non-communicable diseases through increased physical exercise. However, despite the authors using the CHEERS checklist I am unable to formulate an opinion as to whether the evaluation has been conducted appropriately for the following reasons. 1. There is little to no detail provided in the manuscript of the model used to estimate costs and outcomes. Some assumptions are provided in the text, however more detail should be provided albeit in supplementary material to enable the reader to review assumptions, scope and structure of the model.2. The comparator in the evaluation has not been described. For example is it assumed that health outcomes will remain the same over 50 years without the bicycle path investment? Air pollution might be expected to decline with time due to a move away from fossil fuel to electric cars and other environmental initiatives. Community interventions/education and awareness aimed at increasing physical activity might also be expected to lead to better health outcomes with time. Bike use may increase irrespective of the investment in bike paths. Is there evidence of increasing use now? And so on.3. No detail is provided on how DALYs were calculated either for the intervention or the comparator.4. The sensitivity analysis appears to have been limited to the number choosing bikes over cars. No assessment of sensitivity to costs and outcomes has been provided. This is particularly important given the costs have been estimated from a range of sources from the literature and the outcomes have been extrapolated over 50 years.5. Figure 2 a and b are missing scales.
---

	6. It is not possible to see how the CERs were calculated as there is no supporting table. The CER is stated as being the difference between the investment and the health care savings divided by DALYs averted. If this is the case then it does not provide the basis for assessing cost effectiveness. This would require an ICER i.e. the (cost of the intervention – cost of comparator)/(Intervention DALYs – comparator DALYS). An ICER would identify whether the intervention was dominant (i.e. cost saving and better outcomes) or if not whether the investment would be cost effective to achieve the better outcomes in relation to a WTP threshold. Irrespective there is insufficient detail provided to know how the CER was calculated for example what were the DALYs averted – this has not been reported?
--	--

REVIEWER	Mark Stevenson The University of Melbourne Australia
REVIEW RETURNED	01-May-2019

GENERAL COMMENTS	I enjoyed the reading the paper which is a valuable contribution to the literature. Seldom are health economic assessments of this caliber presented; congratulations to the authors. The focus on active transport and infrastructure facilitating such is important and will become increasingly so as we search for safe and sustainable transport in our urban agglomerations. The paper is well written, so the points listed below relate more to aspects of the paper that need clarification and with respect to injury; some cautionary inclusions. 1) The authors refer to a population base comprising linked data from the ASTRID database and the LuTrans database. From what I can glean, the former provides data at an individual level whilst the latter, which relates to traffic flows, is clearly at the aggregate level? On this basis, what do the authors mean when they state they linked the data and how was this used? This aspect of the paper needs clarification. 2) The authors correctly indicated that the change in air pollution overtime would be a disadvantage to additional bicycle riders in relation to health outcomes. However, the authors make no mention as to how long they attributed the length of disadvantage which is clearly subject to the roll-out of the infrastructure? 3) The estimates for injury attributed to bicycling were based on Nilsson et al's estimates. It is important to note that these differ to those advocated by Rune Elviks and it would be useful to include a variety of these approaches in the sensitivity analysis so that one can observe the variation given Stockholm/Sweden's road safety performance is leading the world. 4) There is no mention of the type of cycling infrastructure being developed. If the focus is on infrastructure that separates cyclists from motorised transport, the risk of injury per VKT will be very different which would also affect the estimates reported in your paper. I am aware of a paper that looked at such infrastructure and DALYS associated transitioning to active transport which could assist in applying such an assumption (see Stevenson et al Land use, transport and population health; estimating the health benefits of compact cities. Lancet, 2016; published online Sept 23. http://dx.doi.org/10.1016/S0140-6736(16)30067-X)
---

VERSION 1 – AUTHOR RESPONSE

Reviewer: 1

Reviewer Name: Martin Howell

Institution and Country: University of Sydney Please state any competing interests or state 'None declared': None declared

Please leave your comments for the authors below The economic evaluation addresses a topic of interest and relevance to prevention of non-communicable diseases through increased physical exercise. However, despite the authors using the CHEERS checklist I am unable to formulate an opinion as to whether the evaluation has been conducted appropriately for the following reasons.

1. There is little to no detail provided in the manuscript of the model used to estimate costs and outcomes. Some assumptions are provided in the text, however more detail should be provided albeit in supplementary material to enable the reader to review assumptions, scope and structure of the model.

*Response: The methods used for these calculations have now been described in more detail (page 6, lines 120-127, and page 8 rows 160-166).

2. The comparator in the evaluation has not been described. For example, is it assumed that health outcomes will remain the same over 50 years without the bicycle path investment? Air pollution might be expected to decline with time due to a move away from fossil fuel to electric cars and other environmental initiatives. Community interventions/education and awareness aimed at increasing physical activity might also be expected to lead to better health outcomes with time. Bike use may increase irrespective of the investment in bike paths. Is there evidence of increasing use now? And so on.

*Response: A clarification about the situation to which the scenario is being compared has now been added (page 6, lines 127-128). In terms of exposures this has been clarified in the methods section (page 9, lines 192-193) and also mentioned in the Discussion (page 17, lines 366-370). Due to the uncertainty over changes in exposures during the 50-year calculation period, only changes due to the mode shift from car to bicycle were assessed. Even though vehicle emissions would be expected to change during the modelled time period, there is great uncertainty about their constituents.

3. No detail is provided on how DALYs were calculated either for the intervention or the comparator.

*Response: A more detailed description has now been added to the end of "Estimating health outcomes" (page 8, lines 160-166).

4. The sensitivity analysis appears to have been limited to the number choosing bikes over cars. No assessment of sensitivity to costs and outcomes has been provided. This is particularly important given the costs have been estimated from a range of sources from the literature and the outcomes have been extrapolated over 50 years.

*Response: We agree with the reviewer that the sensitivity to resulting health outcomes (exposure-response functions) and implemented life-time health care cost are important. These have now been added (page 13, lines 249-251; and results added to Figure 3).

5. Figure 2 a and b are missing scales.

*Response: Scales have been added, and we have merged Figure 2 a and b as suggested by the Editor.

6. It is not possible to see how the CERs were calculated as there is no supporting table. The CER is stated as being the difference between the investment and the health care savings divided by DALYs averted. If this is the case then it does not provide the basis for assessing cost effectiveness. This would require an ICER i.e. the (cost of the intervention – cost of comparator)/(Intervention DALYs – comparator DALYS). An ICER would identify whether the intervention was dominant (i.e. cost saving

and better outcomes) or if not whether the investment would be cost effective to achieve the better outcomes in relation to a WTP threshold. Irrespective there is insufficient detail provided to know how the CER was calculated for example what were the DALYs averted – this has not been reported?

*Response: We agree with the reviewer that a table of investment costs, health care costs and DALYs averted would be beneficial. This has now been added (Table 2). Since the cost-effectiveness is calculated only for one intervention (that is only one CER is calculated), we report the CER.

Reviewer: 2

Reviewer Name: Mark Stevenson

Institution and Country: The University of Melbourne Australia Please state any competing interests or state 'None declared': None

Please leave your comments for the authors below I enjoyed the reading the paper which is a valuable contribution to the literature. Seldom are health economic assessments of this caliber presented; congratulations to the authors.

The focus on active transport and infrastructure facilitating such is important and will become increasingly so as we search for safe and sustainable transport in our urban agglomerations. The paper is well written, so the points listed below relate more to aspects of the paper that need clarification and with respect to injury; some cautionary inclusions.

1) The authors refer to a population base comprising linked data from the ASTRID database and the LuTrans database. From what I can glean, the former provides data at an individual level whilst the latter, which relates to traffic flows, is clearly at the aggregate level? On this basis, what do the authors mean when they state they linked the data and how was this used? This aspect of the paper needs clarification.

*Response: The usage of these data sources has now been clarified (page 7, lines 137-138), and in the Abstract (page 2, lines 37-40).

2) The authors correctly indicated that the change in air pollution overtime would be a disadvantage to additional bicycle riders in relation to health outcomes. However, the authors make no mention as to how long they attributed the length of disadvantage which is clearly subject to the roll-out of the infrastructure?

*Response: We agree with the reviewer that there is likely a lag time between exposure and outcome for air pollution. This has been added to the Discussion (pages 18-19, lines 371-375). Physical activity lag times could be found in the literature and were included in the calculations. However, knowledge about lag times for air pollution is limited. The US EPA used segmented lag-times for mortality. Due to lack of studies on lag times for air pollution effects on morbidity, these effects were assumed to occur in the same year as the exposure.

3) The estimates for injury attributed to bicycling were based on Nilsson et al.'s estimates. It is important to note that these differ to those advocated by Rune Elviks and it would be useful to include a variety of these approaches in the sensitivity analysis so that one can observe the variation given Stockholm/Sweden's road safety performance is leading the world.

* Response: We based the estimates for injury attributed to bicycling on the study by Nilsson et al. because their impact evaluation was appropriate for our scenario and study population. A major strength of the Nilsson study, compared to previous health economic assessments of increased bicycling, is their use of national hospital register data on traffic accidents. Since most injuries are not reported to police, other estimates are likely to underestimate injury-related health impacts.

4) There is no mention of the type of cycling infrastructure being developed. If the focus is on infrastructure that separates cyclists from motorised transport, the risk of injury per VKT will be very different which would also affect the estimates reported in your paper. I am aware of a paper that looked at such infrastructure and DALYS associated transitioning to active transport which could assist in applying such an assumption (see Stevenson et al Land use, transport and population

health; estimating the health benefits of compact cities. Lancet, 2016; published online Sept 23. [http://dx.doi.org/10.1016/S0140-6736\(16\)30067-X](http://dx.doi.org/10.1016/S0140-6736(16)30067-X)

*Response: We thank the reviewer for this suggestion. Unfortunately we have no information on specific land-use and types of infrastructure to be built by the City of Stockholm. We have therefore not been able to implement any modification to the current hospital injury data.

VERSION 2 – REVIEW

REVIEWER	Martin Howell University of Sydney Australia
REVIEW RETURNED	21-Jun-2019

GENERAL COMMENTS	The authors have addressed my comments and questions and in so doing have highlighted a problem with the way the CER has been used and confirmed to me that a meaningful cost effectiveness assessment has not been undertaken. The basis for the CER is clearer with Table 1, however usual practice would present costs and outcomes for both the intervention and the comparator and then show the ICER in the table. However, Table 1 suggests that both costs are saved and DALYs averted (i.e. it would be a dominant intervention) and therefore it is not appropriate to calculate an ICER. Irrespective as there is no meaningful comparator the CER does not provide an appropriate basis for evaluating cost effectiveness. The Swedish threshold quoted relates to consideration of ICERs where the costs of an intervention exceed the alternative, however the benefits are greater. The threshold is a willingness to pay value below which the cost required to avert 1 DALY would be considered cost effective. The authors have not conducted a cost effectiveness assessment rather the benefits and costs have been compared to the 2017 baseline burden. A meaningful cost effectiveness analysis would require modelling of both costs and outcomes in the absence of the investment. The baseline 2017 burden of disease will not stay the same over 50 years, improving health is to be expected without the bike paths investment or increased use of bikes, also bike usage could increase and air quality improve all of which needs to be considered in a cost effectiveness analysis. The intervention may still be dominant or have an ICER below the WTP threshold, however this cannot be determined from this assessment. Essentially all that can be stated is that increasing bike use is expected to improve long-term health outcomes and that these would outweigh increased harms from higher traffic injuries and higher air pollution exposure among the additional bike riders. There is an associated decline in DALYs. Given the high costs of chronic illness then it is conceivable that the intervention would be cost effective or even cost saving, however this cannot be determined as the changes in health outcomes and costs if the investment was not made is not known. The manuscript should be revised to reflect the type of assessment that has been undertaken, as in the current form it is not possible to say whether the investment is cost saving or cost effective.
---

REVIEWER	Mark Stevenson University of Melbourne
REVIEW RETURNED	01-Jul-2019

GENERAL COMMENTS	This is a terrific paper now - corrections have been amended and the paper is a valuable contribution
---

VERSION 2 – AUTHOR RESPONSE

Reviewer 1 Comments to Author:

The authors have addressed my comments and questions and in so doing have highlighted a problem with the way the CER has been used and confirmed to me that a meaningful cost effectiveness assessment has not been undertaken.

The basis for the CER is clearer with Table 1, however usual practice would present costs and outcomes for both the intervention and the comparator and then show the ICER in the table. However, Table 1 suggests that both costs are saved and DALYs averted (i.e. it would be a dominant intervention) and therefore it is not appropriate to calculate an ICER. Irrespective as there is no meaningful comparator the CER does not provide an appropriate basis for evaluating cost effectiveness. The Swedish threshold quoted relates to consideration of ICERs where the costs of an intervention exceed the alternative, however the benefits are greater. The threshold is a willingness to pay value below which the cost required to avert 1 DALY would be considered cost effective. The authors have not conducted a cost effectiveness assessment rather the benefits and costs have been compared to the 2017 baseline burden. A meaningful cost effectiveness analysis would require modelling of both costs and outcomes in the absence of the investment. The baseline 2017 burden of disease will not stay the same over 50 years, improving health is to be expected without the bike paths investment or increased use of bikes, also bike usage could increase and air quality improve all of which needs to be considered in a cost effectiveness analysis. The intervention may still be dominant or have an ICER below the WTP threshold, however this cannot be determined from this assessment.

Essentially all that can be stated is that increasing bike use is expected to improve long-term health outcomes and that these would outweigh increased harms from higher traffic injuries and higher air pollution exposure among the additional bike riders. There is an associated decline in DALYs. Given the high costs of chronic illness then it is conceivable that the intervention would be cost effective or even cost saving, however this cannot be determined as the changes in health outcomes and costs if the investment was not made is not known. The manuscript should be revised to reflect the type of assessment that has been undertaken, as in the current form it is not possible to say whether the investment is cost saving or cost effective.

Authors' Response to Reviewer 1 Comments:

We firstly thank the Reviewer for these most important comments. They serve as a reminder that more work needs to be done to develop methods and data to support rigorous economic evaluations of public health interventions.

We agree with the Reviewer that “usual practice” in cost effectiveness analysis has been to present the costs and outcomes for both the intervention and the comparator. We have therefore added also the estimated health care costs based on registry data on current (year 2017) age and gender specific disease incidences.

We also agree that “a meaningful cost effectiveness analysis would require modelling of both costs and outcomes in the absence of the investment”. Ideally the effectiveness of interventions should be assessed by randomised controlled trials. However one of the challenges of undertaking economic evaluations of public health interventions has been the absence of meaningful comparators. In spite of this shortcoming, rigorous economic evaluations of public health interventions and investments are urgently needed to ensure transparency in decision-making. This is also reflected in the second bullet point after the Abstract. We have also added several sentences in the Limitations section to explain why such modelling was not performed. The text follows. Two new references have also been included in our revised paper.

“We acknowledge the limitations in the health economic modelling. It is fair to assume that improvements in population health would still occur without the intervention or increased bicycling. However, modelling all possible behavioural changes under uncertainty and generalising these to population health outcomes in the coming fifty years would be a complex exercise beyond the scope of this study. Furthermore, it would not be meaningful to base predictions on possible future health trends for which there is a lack of reliable data. Such is the case regarding air pollution trends in Stockholm since 2010 [1]. Cost-effectiveness studies facing these and similar limitations have been published previously [2].”

We take the view that it is important to make the best use of available data as we have done here. We also believe strongly in advocating for the development of an evidence-base covering wide ranging population outcomes attributable to public health interventions.

Reviewer 2 Comments to Author:

This is a terrific paper now - corrections have been amended and the paper is a valuable contribution.

Authors’ Response to Reviewer 2 Comments:

We thank the reviewer for this most gratifying feedback.

References

1. Olstrup H, Forsberg B, Orru H, et al. Trends in air pollutants and health impacts in three Swedish cities over the past three decades. *Atmospheric Chemistry and Physics* 2018;18(21):15705–23 doi: <https://doi.org/10.5194/acp-18-15705-2018>[published Online First: Epub Date]].
2. Lindholm L, Stenling A, Norberg M, et al. A cost-effectiveness analysis of a community based CVD program in Sweden based on a retrospective register cohort. , 18(1), 1–7. *BMC Public Health* 2018;18(1):1-7 doi: <https://doi.org/10.1186/s12889-018-5339-3>[published Online First: Epub Date]].

VERSION 3 – REVIEW

REVIEWER	Martin Howell University of Sydney Australia
REVIEW RETURNED	08-Aug-2019

GENERAL COMMENTS	I thank the authors for addressing my concerns and providing sufficient information with respect to assumptions used and the associated limitations and for making changes to the manuscript. I look forward to the final publication.
--